# Efficient Multiple Channels EEG Signal Classification Based on Hierarchical Extreme Learning Machine

**DOI:** 10.3390/s23218976

**Published:** 2023-11-04

**Authors:** Songyang Lyu, Ray C. C. Cheung

**Affiliations:** Department of Electrical Engineering, City University of Hong Kong, Hong Kong SAR, China; yang.lyu@my.cityu.edu.hk

**Keywords:** BCI system, EEG signal, machine learning

## Abstract

The human brain can be seen as one of the most powerful processors in the world, and it has a very complex structure with different kinds of signals for monitoring organics, communicating to neurons, and reacting to different information, which allows large developments in observing human sleeping, revealing diseases, reflecting certain motivations of limbs, and other applications. Relative theory, algorithms, and applications also help us to build brain-computer interface (BCI) systems for different powerful functions. Therefore, we present a fast-reaction framework based on an extreme learning machine (ELM) with multiple layers for the ElectroEncephaloGram (EEG) signals classification in motor imagery, showing the advantages in both accuracy of classification and training speed compared with conventional machine learning methods. The experiments are performed on software with the dataset of BCI Competition II with fast training time and high accuracy. The final average results show an accuracy of 93.90% as well as a reduction of 75% of the training time as compared to conventional deep learning and machine learning algorithms for EEG signal classification, also showing its prospects of the improvement of the performance of the BCI system.

## 1. Introduction

The human brain can be seen as a powerful and comprehensive system responsible for processing, monitoring all behaviors, and reacting to the outside through neural activities, and it has various research aspects in the fields of biology and neuroscience. To bridge the gaps between the human brain and electronic devices, including computers, the brain–computer interface (BCI) system has shown its performance by decoding and processing the signals in the human brain. Therefore, one of the most essential tasks of developing a BCI system is to discover and decipher the various signals generated by the brain and establish their relationship with the corresponding behaviors. However, the collection of the signals in the human brain may be different. With the development of this area, among the signal collection methods, the ElectroEncephaloGram (EEG) signal is widely utilized due to its safety in collection and high resolution in recording, which are its advantages over previous methods. It mainly involves detecting electrical activity in the brain using small metal discs (work as electrodes) attached to the scalp, measuring voltage fluctuations resulting from ionic current within the brain’s neurons [1]. Basically, EEG signals are often divided into different types by frequency, and researchers have already identified and explored various applications and uses for these EEG frequency bands [2]. As summarized in Table 1, five waves correspond to five ranges of frequencies in EEG signals [3].

As for the functions and applications, EEG signals are very powerful tools as a standard in medical and diagnostic research [6], building communication system [7], hand movement direction [8], and driver fatigue detection [9]. To analyze certain EEG signals, big data is growing in the field of healthcare [10], and relative machine learning algorithms improve the efficiency in analysis. Among the applications of EEG signal, motor imagery is the mental execution of a movement without any overt movement or without any peripheral activation [11]. These activities can be reflected by the EEG signal after certain processing.

To realize these functions, BCI systems serve as a connection between the human brain and devices in brain science studies [12]. This technology enables users to control certain electronic devices through the bio-signals generated from brain activities, where the EEG signals or other bio-signals act as the media. About 100 years ago, the relationship between signals and behaviors was first identified by Hans Berger in 1924 when he recorded human brain activity by using the EEG signal. Firstly, Berger could identify the oscillatory activities, such as the alpha wave. Over the years, BCIs have primarily focused on different aspects, such as researching, assisting, or repairing human cognitive or sensory-motor functions [13]. Presently, BCI systems have found broad applications in areas such as sleep monitoring [14], disease detection [15], motor imagery [16], visual realization control, and rehabilitation [17]. Among these applications, motor imagery EEG signals are usually collected from the brain motor cortex area when a person imagines performing specific movements [18]. These EEG signals are typically found in the alpha and central beta frequency bands and can be used to control electronic devices, such as moving a computer’s cursor, operating a wheelchair, or manipulating a robotic arm. Various acquisition techniques can capture motor imagery EEG signals.

To collect EEG signals, this has become easy in recent days, and general EEG devices usually use electrodes, detecting the brain activities by allocating them on the subjects’ scalps consisting of amplifiers, filters, and an analog-to-digital converter. There are portable EEG machines, wireless EEG machines, noninvasive scalp EEG, and other types of machines for specific applications. However, among the BCI applications using EEG signals, it is more difficult to connect the processing and recognition of motor imagery signals and control commands for the computer system than collecting the signals. It may take a lot of time to detect and decode with experiments. Generally, in machine learning algorithms for signal processing, the whole analysis procedure can be separated into two main parts: feature extraction of the signals and the final classifying of the signals. As for feature extraction methods, currently, there are many different algorithms and methods that have been applied including band power (BP) analysis [19], power spectral density (PSD) values [20], and Principal Component Analysis (PCA) [21]. Commonly used classification techniques for interpreting the results include support vector machines (SVM) [22], neural networks-based machine learning methods [23], K-Nearest Neighbors (K-NN) [24], and Naive Bayes [25]. This paper proposes a method that utilizes independent component analysis (ICA) and hierarchical extreme learning machine (H-ELM), combined for motor imagery recognition with the classification of EEG signals. The following are the main contributions of this work:With the powerful learning ability of H-ELM, all-channel signals are processed rather than manually select specific channels for particular purposes, which may be more suitable for different brain–computer interface (BCI) systems of multi-function applications.Optimize and apply hierarchical-ELM (H-ELM) on EEG signal classification and this method can be extracted to all 1-D signals.The results show an accuracy of 93.9% for the classification in motor imagery and an improvement in reducing 75% of the training time of conventional machine learning algorithms.

The following content is proposed as follows: Section 2 illustrates the materials and dataset in this work; Section 3 shows the whole methodology, including the theory, algorithms, and designs in detail; Section 4 presents and compares the results with analysis; and Section 5 concludes the work with future works.

## 2. Materials

The dataset applied and processed in the experiments of this work is offered by the BCI Competition with its Dataset Ia [26] for motor imagery classification based on a 6-channel EEG signal. For applications in the real world, there are different types of EEG signal-collecting devices with different accuracy, channels, and functions. For example, in this work, the data are collected and saved in a 6-channel EEG signal as shown in Figure 1, and for others, it may be saved in a 128-channel EEG. For different applications, some channels may need to be abandoned by the professional for higher efficiency and accuracy. However, in this work, we design a system processing on all channel signals for further applications to make it more general to be used on the BCI system. This can help it be used by different people to avoid further selections. All channels of signals are extended into one sequence for comprehensive analysis. The data can be easily transformed into EEGLab or other methods.

According to the description of the dataset, the data were collected by PsyLab EEG8 amplifier. The A/D-converter utilized was the Computer Boards PCIM-DAS1602/16 bit, with the amplitude of +/−1000 μV for the recorded data. The sampling rate employed during the recording was 256 samples per second (S/s). For the positions of electrodes, the EEG data were collected by 6-channel electrodes with the location set as Channel 1, represented by the A1-Cz position in the 10/20 system, with A1 indicating the left mastoid. Channel 2 corresponded to the A2-Cz position. Channel 3 was located 2 cm frontal to C3, while Channel 4 was positioned 2 cm parietal to C3. Similarly, Channel 5 was situated 2 cm frontal to C4, and Channel 6 was placed 2 cm parietal to C4 [26]. As for details, the total location is shown in shown in Figure 1.

For the details of the experiments of the dataset, according to the demands, each subject intended to manipulate a computer’s cursor on the screen by performing upward and downward movements, during which the machines recorded all of their cortical potentials. Throughout the recording session, the participant received visual feedback on their potentials. Finally, the positive activities of cortical resulted in moving down the cursor on the screen, while negative activities led to moving down. A duration of about 6 s was recorded on each trail. The sampling rate of this experiment is 256 Hz, and the length of the recording is 3.5 s. Finally, 896 samples were recorded for each channel of each trial. After the experiments, 268 trials were recorded as the training set with the labels, and 293 trials were recorded for testing [26].

## 3. Methodology

An overview of the design flow is shown in Figure 2. To make the signals more general in the processing, the signals are sampled under a certain frequency and recorded into digital signals. Then, ICA is utilized for feature extraction noise reduction and separates the data of 6 channels for the independent analysis. To improve the accuracy in certain applications, another ELM of a single layer is used for channel selections of relative and relative channels in different applications under supervision. Then, the selected channels of EEG signals are sent to the training module of H-ELM, in which the hidden nodes are randomly set and are not changed throughout the whole process. Finally, another single-layer ELM works for final decision making to decide whether there is motor imagery. This work can be divided into 2 parts: pre-processing and training. As shown in Figure 2, the two subfigures in the first column show the pre-processing, and the others show the training and final decisions. The details of the input and output in each step are shown as follows.
Input: Collected EEG signalTransform collected EEG signals into digital formats.Extend all channels of signals into a sequenceOutput: EEG Signal sequence in digital formatsInput: EEG signal sequenceSet the parameters in EEGLab.Apply ICA on EEG signals.Reduce the noise of signals.Output: Processed EEG signal after ICAInput: EEG signal after processed by ICASet the parameters of single layer ELM.Decide whether channels are relative or not.Output: Processed EEG signal after selectionInput: Relative EEG signalSet certain parameters in training such as iterative times and calculation accuracy.Train the H-ELM with training data layer by layerCalculate the final parameter β of each layerCalculate the final training results.Record the accuracy and time in both the training and testing stages.Output: Parameters of the last layer

In this work, we can achieve classification and decision making, using the theory and features of H-ELM to process the EEG signals. As for the core component, H-ELM instead of single-layer ELM represents a multi-layer neural network, with enhanced performance in terms of accuracy due to its tighter and more flexible structure, multiple layers, and more number of neurons. At the same time, other relative steps also help improve the performance of this work to make it more feasible for the BCI system. So, the selection and parameters of EEG signals, the usage of ICA, and the theory of ELM are introduced in this section.

### 3.1. EEG Signal Pre-Processing

Firstly, the data used in this work are reshaped into a certain format for further processing by the feature extraction module and channel selection module, to improve the accuracy in training. As for the raw data from the dataset, all of the reactions of the subjects were recorded in a matrix as a generalized EEG signal in voltages. As an example, the size of this matrix of the training set is 268×5377, where the first entry of each column records the class of each trial in 0 or 1, illustrating whether there is motor imagery or not, and the rest of the columns of this matrix contain all 6 channel EEG signals in a row. Then, the label of each subject is eliminated, and the remaining 5376 entries are reshaped into 6 channels according to the description of the dataset for the further selection of relative channels. After the pre-processing of EEG signals, all the data are sent to the feature extraction module of ICA for noise reduction.

### 3.2. ICA for Feature Extraction

ICA is a very important tool in statistics and machine learning. It is widely used to separate the mixed signals into sub-components or independent ones. Sometimes, the signals collected from different electrodes may be mixed, making it hard for further processing. The signal may sometimes be allocated into one channel in the time sequence, leading to computing errors. It has been widely used for optical imaging of neurons, face recognition, communications, and removing artifacts from EEG signals.

Generally, the definition of ICA in the mathematical model, considering a set of observed signals $x=[x_{1},x_{2},\ldots ,x_{n}]$, aims to find a linear transformation represented by a matrix

A such that the source signals of estimation $s=[s_{1},s_{2},\ldots ,s_{n}]$ can be calculated by [27]:(1)s=Ax
where s and x represent n-dimensional column vectors, and the matrix A represents the mixing matrix.

Therefore, the objective of ICA is to calculate the inversion of the mixing matrix W. Then, we can obtain the estimated source signals by:(2)s=Wx
where $W=A^{-1}$.

Among the machine learning algorithms, many results are generated based on statistics analysis for the estimation. Similarly, the ICA algorithm aims to maximize the statistical independence of the estimated source signals by updating the elements of the mixing matrix through iterative calculations.

Based on the theory of ICA, it is employed to extract the main features related to raw EEG signals’ motor imagery to figure out each trial’s features and reduce the noise of raw data as the feature extraction method. This will help simplify the problems in the training process and improve the training speed.

### 3.3. ELM Theory

As for the core algorithm and its theory in this work, Huang et al. first presented the concept and theory of ELM in 2004. This algorithm and network largely help reduce the training time of networks and improve the performance [28]. Previously, to obtain the results from networks, back-propagation (BP) [19] was widely used by randomly setting the parameters in hidden layers and using the iterative algorithm to update, costing a large number of resources and time. Certain applications may require the network to finish the training process quickly and put it into use in a short time. And the robustness of the network and the frequency of upgrading are also very important. So, ELM shows its advantages in this area because it randomly generates the bias and weights, and once fixed, they do not need to be changed. Then, the results are calculated by the least-square algorithm, making the process more general for all kinds of platforms and coding languages. With less training time, it improves the flexibility for specific applications in realizing frequent training with the latest dataset. The general structure of ELM is presented in Figure 3. For its general calculation process, *X* is defined as the training set. Here, several *N* labeled pairs $\left(x_{i},y_{i}\right)$ are set. $x_{i}\in R$ denotes the *i*th input vector, and $y_{i}\in R$ represents the target. Overall, to describe the ELM function, f(x) is:(3)$$f\left(x\right)=\sum_{i=1}^{M}w_{j}\;{}_{▪}\;a\left(r_{j}\;{}_{▪}\;x+b_{j}\right)$$

For each layer of ELM, there are M neurons of the input layer connecting to *H* neurons in the hidden layer with the weights represented by $r_{j}\in R,j=1,\ldots ,H$. Each hidden neuron, denoted by *j*, incorporates a bias term $b_{j}$ and a nonlinear activation function $a({}_{▪})$. The hidden layer is further connected to the output neurons through a vector defined as w∈R. As for the core feature of ELM, the pairs $r_{j},b_{j}$ in Equation (Equation 1) are randomly defined and not to be tuned in training. Then, let *H* also represent the matrix of activation, where $h_{ij}\in H\left(i=1,\ldots ,N;j=1,\ldots ,H\right)$ represents the *j*th hidden neuron for the *i*th input pattern activation value. The relationship is $h_{ij}=a\left(r_{j}\;{}_{▪}\;x_{i}+b_{j}\right)$, and the detailed results are shown in Equation (Equation 2):(4)$$H=H(w_{1},\ldots ,w_{\tilde{n}},b_{1},\ldots ,b_{\tilde{n}},x_{1},\ldots ,x_{\tilde{n}})$$
(5)$$H=\left[\begin{array}{ccc} a(w_{i}{\dot{x}}_{j}+b_{i}) & \cdots & a(w_{\tilde{N}}\dot{x_{1}}+b_{\tilde{N}})\\ \vdots & \cdots & \vdots \\ a(w_{i}{\dot{x}}_{N}+b_{i}) & \cdots & a(w_{\tilde{N}}\dot{x_{N}}+b_{\tilde{N}}) \end{array}\right]$$

Therefore, ELM training can be simplified in minimizing:(6)$$\min_{\left(w,b\right)}{∥\begin{array}{c} Hw-Y \end{array}∥}^{2}$$

Overall, training an ELM involves a straightforward procedure that can be divided into the following steps:Set the parameters pairs $r_{j},b_{j}$ for the hidden layer randomly;Calculate *H* as the activation matrix;Calculate the output weights by Equation (Equation 3);Finally, the core computation can be expressed as shown in Equation (Equation 5).

### 3.4. Basic Structure of H-ELM

For the disadvantages of conventional ELMs with a single layer, the learning ability may be limited, even if the layer’s scale is very large. The reason is that ELM involves matrix inversion, and the value of the entries of hidden layer varies from −1 to 1. When a certain entry has a very small value, the inversion calculation may lead to a very large output as an outlier, and this will cause inaccurate results in training. To keep the same total number of hidden nodes, separating a large layer into more small layers can not only improve the accuracy but it can also improve the flexibility in setting the layers. Therefore, multi-layer ELM is required for higher accuracy and stable structures.

H-ELM [29] is a framework with multiple layers designed as an effective and efficient Multilayer Perceptron (MLP); with a multi-layer structure, certain requirements including accuracy can be fulfilled compared to conventional ELM. It has shown advantages in the accuracy of 99.13% in the conventional MNIST dataset, where some of the deep learning methods, such as DBM and DBN, only realize accuracies of 99.05% and 98.87%, but deep learning methods cost nearly 200 times to 600 times the consumption in training time [29]. H-ELM separates the machine learning process into two steps: unsupervised multi-layer training, and supervised feature classification and final decision making. To improve the performance in training accuracy and speed, ELM sparse auto-encoder is also employed in H-ELM [30]. In the ELM sparse auto-encoder approach, each layer in the stack architecture operates as an independent subsystem or submodule. Additionally, a random matrix is employed to disperse the high-level features of the networks during feature classification. Finally, the original single-layer ELM is applied for the ultimate decision-making process. The general structure of H-ELM is shown in Figure 4.

In order to establish connections between each single-layer ELM, similar to conventional multi-layer models, the output and weights of the last layer are directly transmitted as input to the subsequent layer. Consequently, an N-layer unsupervised learning process is conducted to obtain high-level sparse features. The calculation of the output for each hidden layer can be expressed as:(7)$$H_{i}=g\left(H_{i-1}\dot{\beta }\right)$$
where $H_{i}$ is the result of the *i*th layer (i∈[1,K]), and $H_{i-1}$ represents the result of the (i−1)th layer respectively. Then g(·) represents the activation function, and β denotes the output weights. Each hidden layer is seen as an independent single-layer ELM in the analysis. The core feature of the ELM structure is that once the hidden layer from the previous layer is randomly set, the weights or parameters of the current hidden layer remain fixed and do not necessitate tuning. This distinguishes H-ELM from deep learning algorithms that utilize backpropagation (BP) calculations and give the high training speed of ELM. Therefore, it reduces the process of setting and revising the hidden layers and improves the overall training speed or time consumption in applications.

As shown in Figure 4, the calculation results of the *i*th layer can be regarded as the input features. In the subsequent step, when these features are utilized in applications, they are randomly set and used to compute the results. This exemplifies the high capability of each layer of ELM in various applications, making it a versatile approach. Similarly, H-ELM can also be summarized in the following steps:Given the training set $N=\left(x_{i},t_{i}\right)|x_{i}\in R^{n},t_{i}\in R^{m},i=1,\ldots ,N$ with the activation function g(x), and hidden neuron number $\hat{N}$.Calculate the hidden layer output matrix *H* by
(8)$$\min_{\left(w,b\right)}{∥\begin{array}{c} Hw-Y \end{array}∥}^{2}+\lambda {∥\begin{array}{c} w \end{array}∥}^{2}$$Obtain β by $\beta =H^{†}T$ as the output.Follow the principle of multi-layer connection to connect each layer through $H_{i}=g\left(H_{i-1}\dot{\beta }\right)$.

### 3.5. ELM Sparse Auto-Encoder

For a further introduction of the ELM sparse auto-encoder, it transfers certain calculations of input data into approximation. In theory, it aims to achieve the equation:(9)$$h_{\theta }\left(x\right)\approx x$$
where θ = A,b and *A* represents the hidden weight, and *b* denotes the bias of the framework. Moreover, the auto-encoder helps reduce errors. Generally, $L_{1}$ optimization is used for establishing the ELM auto-encoder, and the singular values of $L_{2}-norm$ are calculated for high-level features optimization. The optimized mathematics model of this auto-encoder can be written as follows:(10)$$O_{\beta }=\underset{\beta }{\mathrm{argmin}}\left\{{\left||H\beta -X|\right|}^{2}+{\left||\beta |\right|}_{\ell_{1}}\right\}$$

Similarly, *X* denotes the input, *H* represents the randomly mapped output, and β represents the weight. Furthermore, *X* represents the original data for ELM, and *H* is randomly generalized, and once fixed does not need to be tuned. The algorithm for solving H-ELM through a sparse autoencoder is shown as follows:Input: data $X=x_{i}$ where i=1,…,N, output matrix of hidden nodes $H=h_{i}$ where i=1,…,L.Determine the Lipschitz constant (γ) of the gradient of a smooth convex function ∇p.Calculate β iteratively, $y_{1}=\beta_{0}$, $t_{1}=1,i\geq 1$.$\beta_{j}=s_{\gamma }\left(y_{j}\right)$, where $s_{\gamma }$ is given by
(11)sγ=argminβγ2||β−(βi−1−1γ∇p(β(i−1))||2+q(β)γ2||β−(βi−1−1γ∇p(βi−1)||2.$t_{i+1}=\frac{1+\sqrt{1+4t_{i}^{2}}}{2}$.$y_{i+1}=\beta_{i}+\left(\frac{t_{i-1}}{t_{i+1}}\right)\left(\beta_{i}-\beta_{i-1}\right)$.Output weight β.

Using the auto-encoder enhances the computational efficiency and application performance. In the H-ELM framework, the learning process in previous layers is designed for training, while the final layer is responsible for decision making. In this study, after the experiments under various conditions, a three-layer H-ELM is employed in applications with the sparse ELM autoencoder to enhance efficiency. Details of the experiment result with analysis are shown in Section 4.

## 4. Experiment Results and Discussion

In this work, all experiments are performed on a PC with the CPU of Intel(R) Core(TM) i5-6400 CPU @2.70 GHz and 24 GB 2133 MHz RAM. The programs are performed by Matlab 2020a. The data from each subject are labeled as the description of the dataset from [31]. Among the datasets used in our experiments, 268 trials from the training set with labels are used for training, and the other 293 trials are also labeled for testing. For the data from each subject, −1 or 1 are labeled to represent the motor imagery in the test. The results of the experiments are shown in detail as follows.

Firstly, the number of layers should be selected according to this dataset. After primary experiments, under the same conditions, 3-layer H-ELM can realize a balance between the accuracy of classification and time consumption in training as shown in Table 2. To explain the results, the H-ELM may have overfittings in training, and the 3-layer framework can realize the classification. Different numbers of neurons are selected for the EEG signal classification to test and find the most suitable results in balancing the accuracy and training time. In total, the trends in the training process and the testing process are shown in Figure 5 and Figure 6. From the figures, the training accuracy arises 100% if more than 700 neurons are set for training. Therefore, only more than 700 hidden nodes should be considered in this application; otherwise, the wrong training may also lead to wrong testing and final decisions. The training time increases nearly linearly in the training part, so selecting the number of hidden nodes should consider the results in both aspects.

For the testing part, the testing time is also increasing linearly, and it may affect the applications. Therefore, the time in testing may also be influenced by the dataset. However, the trend of testing accuracy is highly related to the number of hidden nodes, which increases when the number ranges from 300 to 1000, and then decreases after the number of hidden nodes is larger than 1000, but it will be nearly unchanged. Besides the overfitting, which is mainly because when the ELM has a very large scale in the hidden matrix, it may lead to unstable results in calculations since inversion calculations are involved in ELM as mentioned in Section 3.4, although the multi-layer structure has separated the errors to more layers. Therefore, the number 1000 is selected as the best performance in this application of EEG signal classification, although it varies depending on the scale of the selected dataset.

Figure 5 and Figure 6 show that H-ELM in EEG signal classification can finish the training within only 2.5 s for this dataset, and the testing can also be finished with 0.04 s. At the same time, the framework shows an accuracy of 93.9% in EEG signal classification based on all channel signals. The results demonstrate the advantages of the proposed approach in terms of both accuracy and training time when compared to conventional machine learning methods and single-layer ELM. The comparison results mainly focus on time consumption in training and testing accuracy, and they are shown in Table 3. From the comparisons with other methods such as K-NN, it has relatively high accuracy, especially in applications with large datasets, but it requires large time consumption. For certain applications that may require fast reactions, these methods may not be able to realize real-time reactions on certain occasions. For certain application areas, such as clinical, the frameworks may need frequent updates to maintain high accuracy. So, the proposed method balances the advantages in training time and accuracy, and it has a very small structure and is easy to be used and applied on hardware [32]. In addition, these experiments do not require the manual selection of the channels, which is more general in application, especially implemented on large systems, such as BCI systems.

## 5. Conclusions

This paper shows the application of H-ELM on the classification of all channel EEG signals with very high training and testing speed compared with other methods, such as improving nearly four times with k-NN for the same application and dataset. The results from this work also show the application prospects in other 1-D signal processing and classifications. Additionally, it can realize an average accuracy of 93.9%. To further improve the accuracy, fast feature extraction methods can be applied to this network, and hidden nodes can also be set for certain applications for proper performance, under the condition that the device can react to the change of the EEG signal from the electrodes in a short time. From these results, H-ELM shows potential for hardware architecture due to its simple structure and low resource occupation. However, ELM may lead to unstable results in the training process because when the scale of the hidden matrix is large, the inversion calculation may have an outlier value in the matrix and cause wrong results. Therefore, the training and testing need to be optimized to obtain the average results. Currently, it shows advantages in some work, such as direction-of-arrival (DOA) measurement of mice EEG [36], emotion detection [37], and building a more flexible BCI system [38]. Therefore, for future research directions, other relative methods, including feature extractions and noise reductions, will be applied, and it can be applied on hardware platforms to realize low-power and real-time measurements and analyses with high and efficient performance.

## Figures and Tables

**Figure 1 sensors-23-08976-f001:**
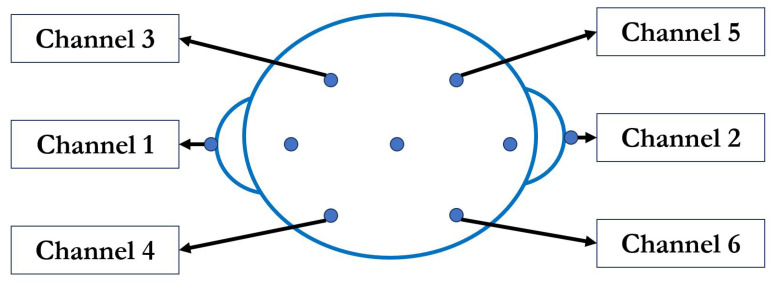
The 6-channel electrodes for EEG signals.

**Figure 2 sensors-23-08976-f002:**
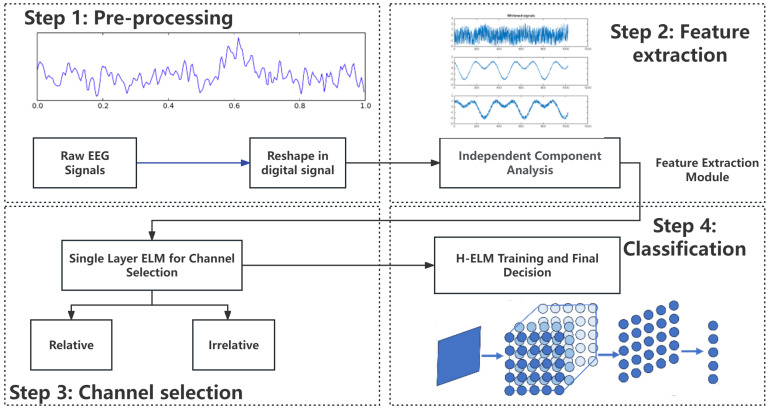
Overview of the design of the whole process.

**Figure 3 sensors-23-08976-f003:**
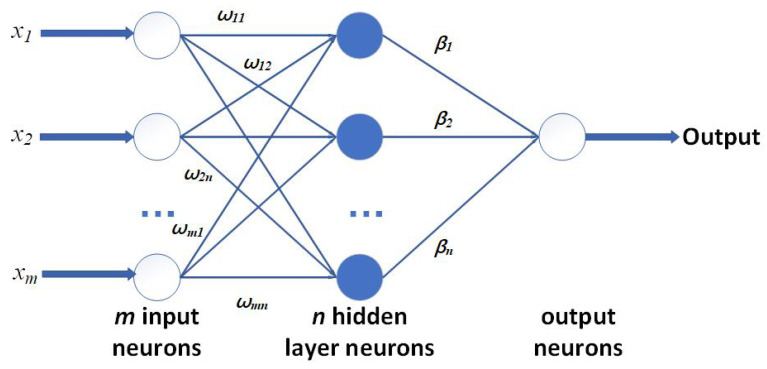
General structure of ELM.

**Figure 4 sensors-23-08976-f004:**
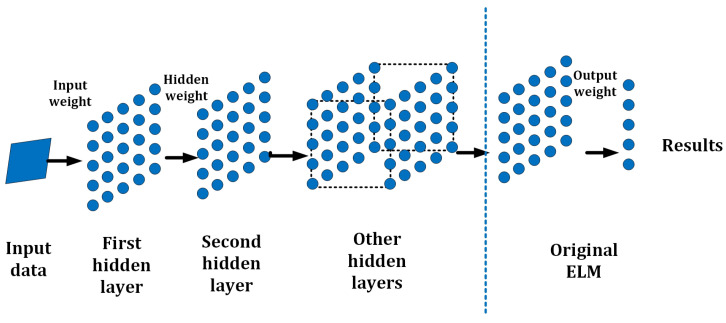
General structure of H-ELM.

**Figure 5 sensors-23-08976-f005:**
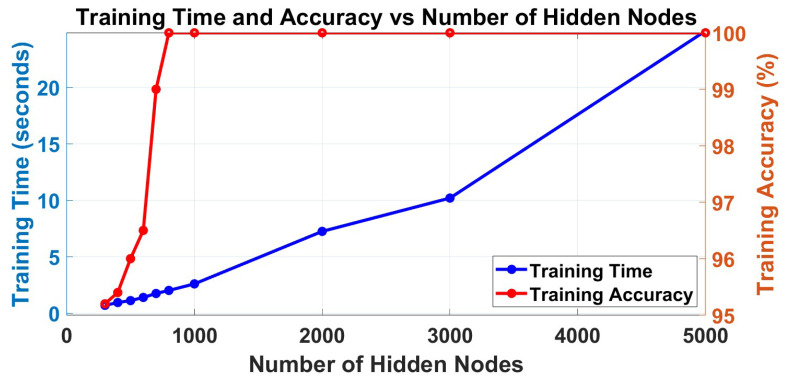
Trend of training accuracy and training time under different numbers of neurons.

**Figure 6 sensors-23-08976-f006:**
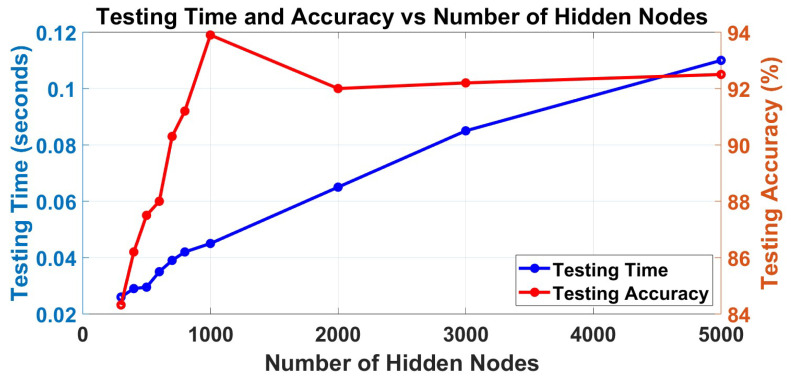
Trend of testing accuracy and testing time under different numbers of neurons.

**Table 1 sensors-23-08976-t001:** Different kinds of EEG signals.

Band	Frequency	Occasions
Delta	0.5–4 Hz	Mainly found in babies, adult slow-wave sleep, and some continuous-attention tasks [4]
Theta	4–8 Hz	Mainly found in young people and children, drowsiness in adults and teens with inhibition of elicited responses [4]
Alpha	8–13 Hz	Found in relaxed, seemingly with the purpose of timing inhibitory activity in different locations across the brain.
Beta	13–30 Hz	Found in active thinking, focus, and anxious
Gamma	larger than 30 Hz	Found in advanced cognitive processing and the merging of sensory information [5].

**Table 2 sensors-23-08976-t002:** Relative results under different numbers of layers.

Layers	Hidden Nodes of Each Layer	Test Accuracy (Average)	Training Time (Average)
3	700	93.90%	1.59 s
4	700	93.25%	2.51 s

**Table 3 sensors-23-08976-t003:** Accuracy and training time of different methods on BCI competition datasets.

Method	Accuracy in Percent	Training Time in Average
ELM	87.50	0.90 s
DBM	93.99	around 100 s
k-NN [33]	92.15	around 10 s
Neural Network [34]	91.47	NA
Bayes [35]	90.44	NA
Proposed H-ELM	93.90	2.60 s

## Data Availability

This paper uses the public dataset of BCI Competition II [26].

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
