# Peer review of "Efficient Multiple Channels EEG Signal Classification Based on Hierarchical Extreme Learning Machine"

_sensors, 2023, doi:10.3390/s23218976_

Round 1
Reviewer 1 Report
Comments and Suggestions for Authors
The author presents an efficient method in EEG signal classification and shows improvement mainly in training speed. However, the following questions should be answered in the revised paper.
1. In machine learning and deep learning areas, different platforms, devices, and algorithm designs may have a significant influence on the training, and then the testing and application results. Did you use the same training set to evaluate compared deep networks in Table 3? Are all the tests performed on the same platform?
2. Every algorithm has shortcomings. What is it in your design? There are many existing studies for EEG signal classifications. Please also illustrate your design’s disadvantages, and put them in the conclusion.
3. MLP and ELM are used widely for bio-medical applications. For these applications, more literature reviews should be added. For example, “Trends in Compressive Sensing for EEG Signal Processing Applications”, “Developing a Deep Neural Network for driver fatigue detection using EEG signals based on Compressed Sensing”, and “Electroencephalography Signal Processing: A Comprehensive Review and Analysis of Methods and Techniques” Cite these papers and some other as related works.
4. The Pre-processing phase of the EEG signal should be described clearly with technical details. What method do you use in reshaping EEG signals? Please illustrate the whole process with clear presentations.
5. The formats of equations and the definitions should be improved.
6. If the method mentioned in this paper can be used in clinical analysis or BCI system? If not, what are the difficulties? You have said a lot about the BCI system, please also supplement certain related works of this.
Reviewer 2 Report
Comments and Suggestions for Authors
The authors propose a fast-reaction framework based on a multi-layer extreme learning machine (ELM) is utilized for the Electroencephalogram (EEG) signals classification in motor imagery. They claimed that the proposed framework achieved high accuracy and less training time compared with conventional machine learning methods. However, the exiting method already achieved higher accuracy (Table 3) as compared to the proposed method and training time depends on the hardware on which computer the experiment is done. Moreover, I have some observations as follows:
· In the abstract, briefly introduce the work and highlight the motivation and contributions, and at the end, highlight the results. There is no need for a lot of descriptions.
· Could you elaborate on the specific characteristics of the multi-layer extreme learning machine (ELM) utilized in your framework, and how it contributes to the improved accuracy and training time in EEG signal classification compared to conventional methods?
· Improve the quality of Figure 1. There is no labeling for x and y axis in Figure 2. So, it is meaningless.
· How does the adjustment of hidden nodes impact the overall accuracy in EEG signal classification? Could you elaborate on the implications of these adjustments in terms of real-time responsiveness and adaptability to dynamic changes in EEG signals?
· How do you ensure the robustness and reliability of the system when dealing with noisy EEG signals or variations in signal quality, and what measures are taken to prevent potential overfitting or underfitting issues in the classification process?
· There are several typos that need to be fixed carefully. For example, “… Support Vector Machines (SVM)[18] …”, “ … the Hidden layer …”, “hidden weight,b is” etc.
· The authors consider only one dataset to evaluate their model. To prove robustness of the proposed model they could consider other datasets too.
· It is a good investigation to see the average recognition time for the proposed method and existing methods. Authors should add both average training and testing/recognition time.
Comments on the Quality of English LanguageMinor editing of English language required
Reviewer 3 Report
Comments and Suggestions for Authors
1. In the introduction part the researchers said (the alpha wave (8-13 Hz)), then after four lines they said (the Alpha (8-12 Hz) ), the Alpha is 8 to 12 or 13,
2. The same thing with Beta, in the table I they said above 14 Hz is Beta while they said (and central Beta (16-25 Hz) frequency bands ), so, Beta is above 14 or 16
3. In the introduction part there is not enough explanation about motor imagery signals
4. I recommended adding a paragraph explaining the EEG devices in the introduction part
5. The material part needs more explanation spatially the BCI Competition II Dataset, how many subjects, frequency rate etc.
6. The sub-section (EEG Signal Pre-processing) is wrong because all the paragraphs under this section talk about the EEG device, the places of electrodes, sampling rate etc. there is no relationship between those sentences and pre-processing methods
7. As I understand from your study there is no pre-processing step, just the signals are converted into a digital number
8. Section 3.2. ICA for feature extraction needs more explanation about the ICA algorithm
9. I think that the first paragraph of section 4 is repeated sentences, you explained it in the third section
The similarity rate of this study is 29%. This is not an acceptable rate.
Author Response
Thank you for your comments. Please refer to the attachment.

Round 2
Reviewer 2 Report
Comments and Suggestions for Authors
This manuscript has been revised according to the comments. The authors have significantly improved the quality of the paper.
Reviewer 3 Report
Comments and Suggestions for Authors
1. At the introduction part the researchers said (the alpha wave (8-13 Hz)), then after four lines they said (the Alpha (8-12 Hz) ), the Alpha is 8 to 12 or 13,
2. The same thing with Beta, in the table I they said above 14 Hz is Beta while they said (and central Beta (16-25 Hz) frequency bands ), so, Beta is above 14 or 16
They Solved all frequencies bans in the Table 1
3. At the introduction part there is not enough explanation about motor imagery signals
They Solved it by Adding:
Among these applications, motor imagery EEG signals are usually collected from the brain motor cortex area when a person imagines performing specific movements [18]. These EEG signals are typically found in the alpha and central beta frequency bands and can be used to control electronic devices, such as moving a computer’s cursor, operating a wheelchair, or manipulating a robotic arm. Various acquisition techniques can capture motor imagery EEG signals.
4. I recommended to add a paragraph explain the EEG devices at the introduction part
They Solved it by Adding:
To collect EEG signals, this has become easy in recent days, and general EEG devices usually use electrodes detecting the brain activities by allocating them on the subjects’ scalps consisting of amplifiers, filters, and an analog-to-digital converter. There are portable EEG machines, wireless EEG machines, noninvasive scalp EEG, and other types of machines for specific applications. However, among the BCI applications using EEG signals, it is more difficult to connect the processing and recognition of motor imagery signals and control commands for the computer system than collecting the signals. It may take a lot of time to detect and decode with experiments.
5. The material part needs more explanation spatially the BCI Competition II Dataset, how many subjects, frequency rate etc.
They Solved it by Adding:
According to the description of the dataset, the data was collected by PsyLab EEG8 amplifier. The A/D-converter utilized was the Computer Boards PCIM-DAS1602/16 bit, with the amplitude of +/-1000 μV for the recorded data. The sampling rate employed during the recording was 256 samples per second (S/s). For the positions of electrodes, the EEG data was collected by 6-channel electrodes with the location as Channel 1 represented the A1-Cz position in the 10/20 system, with A1 indicating the left mastoid. Channel 2 corresponded to the A2-Cz position. Channel 3 was located 2 cm frontal to C3, while Channel 4 was positioned 2 cm parietal to C3. Similarly, Channel 5 was situated 2 cm frontal to C4, and Channel 6 was placed 2 cm parietal to C4[26]. As for details, the total location is shown in shown in Fig. 1.
For the details of the experiments of the dataset, according to the demands, each subject intended to manipulate a computer’s cursor on the screen by performing upward and downward movements, during which the machines recorded all of their cortical potentials. Throughout the recording session, the participant received visual feedback on their potentials. Finally, positive activities of cortical resulted in moving down the cursor on the screen, while negative activities led to moving down. A duration of about 6 seconds was recorded on each trail. The sampling rate of this experiment is 256 Hz, and the length of the recording is 3.5s. Finally, 896 samples were recorded for each channel of each trial.
6. The sub section (EEG Signal Pre-processing) is wrong because all the paragraphs under this section talk about the EEG device, the places of electrodes, sampling rate etc. where is no relationship between those sentences and pre-processing methods
They Solved it by adding:
Firstly, the data used in this work will be reshaped into a certain format for further processing by the feature extraction module and channel selection module, to improve the accuracy in training. As for the raw data from the dataset, all of the reactions of subjects were recorded in a matrix as a generalized EEG signal in voltages. As an example, the size of this matrix of the training set is 268 × 5377, where the first entry of each column recorded the class of each trial in 0 or 1, illustrating if there is motor imagery or not, and the rest of the columns of this matrix contain the all of the 6 channel EEG signals in a row. Then, the label of each subject will be eliminated, and the remaining 5376 entries will be reshaped into 6 channels according to the description of the dataset for further selection of relative channels. After the pre-processing of EEG signals, all the data will be sent to the feature extraction module of ICA for noise reduction.
7. As I understand from your study there is no pre-processing step, just the signals are converted into a digital number
After rewriting the above paragraph in comment No. 6…..now I understand the preprocessing step
8. Section 3.2. ICA for feature extraction needs more explanation about ICA algorithm
They Explained it
ICA is a very important tool in statistics and machine learning. It is widely used to separate the mixed signals into sub-components or independent ones. Sometimes, the signals collected from different electrodes may be mixed making it hard for further processing. Besides, the signal may sometimes be allocated into one channel in the time sequence, leading to computing errors. It has been widely used for optical imaging of neurons, face recognition, communications, and removing artifacts from EEG signals. Generally the definition of ICA in the mathematical model, considering a set of observed signals x = [x1, x2, . . . , xn], aims to find a linear transformation represented by a matrix A such that the source signals of estimation s = [s1, s2, . . . , sn] can be calculated by[27]:
9. I think that the first paragraph of the section 4 is repeated sentences, you explained it at the third section
Now Section 4 has started in a good way
Comments on the Quality of English LanguageMinor editing of English language required